# Climate Precursors of Satellite Water Marker Index for Spring Cholera Outbreak in Northern Bay of Bengal Coastal Regions

**DOI:** 10.3390/ijerph181910201

**Published:** 2021-09-28

**Authors:** Tomomichi Ogata, Marie-Fanny Racault, Masami Nonaka, Swadhin Behera

**Affiliations:** 1Japan Agency for Marine-Earth Science and Technology, Yokohama 237-0061, Kanagawa, Japan; nona@jamstec.go.jp (M.N.); behera@jamstec.go.jp (S.B.); 2Plymouth Marine Laboratory (PML), Prospect Place, The Hoe, Plymouth PL1 3DH, UK; mfrt@pml.ac.uk; 3National Centre for Earth Observation (NCEO), Prospect Place, The Hoe, PML, Plymouth PL1 3DH, UK

**Keywords:** cholera, tropics, remote sensing, climate variability

## Abstract

Cholera is a water-borne infectious disease that affects 1.3 to 4 million people, with 21,000 to 143,000 reported fatalities each year worldwide. Outbreaks are devastating to affected communities and their prospects for development. The key to support preparedness and public health response is the ability to forecast cholera outbreaks with sufficient lead time. How *Vibrio cholerae* survives in the environment outside a human host is an important route of disease transmission. Thus, identifying the environmental and climate drivers of these pathogens is highly desirable. Here, we elucidate for the first time a mechanistic link between climate variability and cholera (Satellite Water Marker; SWM) index in the Bengal Delta, which allows us to predict cholera outbreaks up to two seasons earlier. High values of the SWM index in fall were associated with above-normal summer monsoon rainfalls over northern India. In turn, these correlated with the La Niña climate pattern that was traced back to the summer monsoon and previous spring seasons. We present a new multi-linear regression model that can explain 50% of the SWM variability over the Bengal Delta based on the relationship with climatic indices of the El Niño Southern Oscillation, Indian Ocean Dipole, and summer monsoon rainfall during the decades 1997–2016. Interestingly, we further found that these relationships were non-stationary over the multi-decadal period 1948–2018. These results bear novel implications for developing outbreak-risk forecasts, demonstrating a crucial need to account for multi-decadal variations in climate interactions and underscoring to better understand how the south Asian summer monsoon responds to climate variability.

## 1. Introduction

The northeast Indian Ocean is acknowledged to be the cradle of the bacterial pathogen *Vibrio cholerae* implicated in the current pandemic of cholera [1,2]. This life-threatening water-borne disease represents a significant global health threat, affecting [3]. 3 to 4 million people each year worldwide, with 21,000 to 143,000 reported fatalities. The regions around the northern Indian Ocean accounted for 57% of the total number of reported cholera cases in 2016 and 43% in the period 2010–2016 (World Health Organization, Global Health Observatory (GHO) data), with the highest incidence of cases reported in densely populated coastal areas from contaminated drinking water and seafood sources, and through bathing and recreational of contaminated water bodies [4].

Environmentally favorable habitats for water-borne pathogens are expanding under climate change [5], and specifically, the habitat suitability of *V. cholerae* is estimated to have increased by ≈10% globally compared with a 1980s baseline [6]. The distribution of *V. cholerae* population and transmission routes of cholera disease have been shown to be influenced by extreme climate and weather events such as droughts, floods, and storm surges. For instance, during three flood-associated epidemics (1988, 1998, and 2004) in Dhaka in Bangladesh, the pathogenic bacteria *V. cholerae* was the most commonly identified responsible agent of acute diarrheal disease [7]. Particularly, in 1998, the severe flood was associated with further environmental changes in the Bengal Delta coastal waters in the north of the northern Bay of Bengal (BoB), including the warming of sea surface temperature (SST), increased chlorophyll-a concentration, and turbidity associated with enhanced river discharge and sewage, which are recognized as favorable growth conditions for the bacteria [8,9,10,11,12]. These changes in environmental conditions, compounded by the limited access to safe drinking water and poor sanitary conditions, led to a marked increase of cholera incidences and with 25% of 3109 people surveyed in two rural areas of Bangladesh reporting severe diarrheal illness [13].

The seasonality of cholera outbreaks in the Bengal Delta region shows semiannual cycles characterized by one peak in boreal spring and the other in boreal fall [14,15]. The major monitoring program of cholera disease incidence, such as the International Centre for Diarrheal Disease Research in Bangladesh (ICDDR, B) and the Integrated Disease Surveillance Programme (IDSP) in India, have demonstrated their outstanding importance for epidemiological research [15,16,17,18], but they also noted some limitations. Epidemiological time-series of cholera are typically noisy, complex, and strongly non-stationary [16], and the datasets may remain limited in space and/or time. In addition, disease outbreaks may be under- or mis-reported due to insufficient laboratory with adequate testing capacity, lack of staff training to disease-case definitions often resulting in cholera cases being reported as ‘acute diarrheal disease’ or patients not seeking hospital treatment if symptoms are mild [19,20,21].

Over the past decades, research studies have revealed large-scale and regional climate variability such as the El Niño/Southern Oscillation (ENSO), Indian Ocean Dipole (IOD), and south Asian summer monsoon rainfall to alter local and regional temperature and precipitation, creating conditions favorable for *V. cholerae* growth and increasing the risk of cholera outbreaks [22,23,24,25,26]. In particular, short-term variation in cholera transmission rates in Bangladesh showed a significant 8–10 month lagged correlation with the Niño3.4 ENSO index [27]. Similarly, in regions under the monsoon influence, such as India and Bangladesh, the anomalous SST warming in the tropical Pacific following winter El Niño events and the related changes in summer monsoon atmospheric circulation have been associated with enhanced precipitation and river discharges over those regions, leading to a rise in cholera outbreaks [28,29]. However, mechanistic understanding of the dynamics of the climate drivers of cholera outbreaks have remained limited, and particularly the relationship between climate drivers and changes in biological characteristics in the coastal environment of BoB for promoting *V. cholerae* bacteria are yet to be elucidated.

In the cholera-endemic region of northern BoB, a Satellite Water Marker has been developed to characterize the biological condition of coastal waters and their potential suitability for *V. cholerae* bacteria [3]. The SWM is estimated based on the variability difference between remote-sensing reflectance in the blue (412 nm) and green (555 nm) wavelengths. Interestingly, although remote-sensing chlorophyll-a concentration has been demonstrated as a useful predictor for environmental reservoirs of *V. cholerae* bacteria [17,18,22,30] and further comparative analysis along northern Indian coastal regions showed high-positive correlations (R^2^ > 80%) between SWM and chlorophyll-a [11], the SWM demonstrated higher predictive ability (R^2^ = 78%) compared to chlorophyll-a concentration (R^2^ = 58%) for the prediction of spring season cholera incidence in the Bangladesh region [3].

Using remote sensing and numerical simulations, recent studies have revealed the relationship between biological oceanography and climate variability. For example, previous studies [31,32] showed that the high chlorophyll-a concentration (bloom) occurs along the Sumatra–Java coast during positive IOD years through strong coastal upwelling. On the other hand, biological condition over the northern BoB is expected to be controlled by not only coastal upwelling but also river discharge from the Ganges and Brahmaputra rivers. Therefore, Indian monsoon rainfall variability over these river basins is important for the chlorophyll-a concentration and SWM dynamics. A recent study [33] further demonstrates a 2-year prediction skill for Indian summer monsoon rainfall controlled by the multi-year oceanic memory such as ENSO. These studies suggest that the climate variability significantly contributes to the coastal biological condition, and it may be predictable beyond the season (or multi-year) using accurate climate prediction [31,33].

Here, we analyze a new two-decade long SWM time-series based on satellite climate-quality data records, to identify, for the first time, the climate precursors of biological oceanic conditions favorable for *V. cholerae* and seasonal cholera outbreaks in the northern BoB coastal regions. First, we use the summer monsoon rainfall index over the northern India region to investigate the climate precursor for the potential driving factor (i.e., summer monsoon flooding) of the SWM index that is related to spring cholera outbreak. As possible climate drivers to the SWM index over the northern BoB coastal region, we consider global patterns of SST that drive regional changes in surface temperature and rainfall over northern India and river runoff over the Bengal Delta region through the climate mode teleconnection. Finally, we revisit the decadal change of the ENSO–monsoon relationship from the perspective of application to long-term forecast of cholera risk.

## 2. Materials and Methods

### 2.1. Climate and River Basins Data

In this study, we have used high-resolution daily rainfall data of the Indian Meteorological Department (IMD) [34]. The IMD data are at a resolution of 0.25° × 0.25° over the period 1901 to present (we used 1948–2017) available at: https://www.imdpune.gov.in/Clim_Pred_LRF_New/Grided_Data_Download.html.

Monthly composites are calculated, and the interannual anomalies for these monthly climate fields are derived by removing monthly climatology from their monthly values. The gridded monthly SST anomalies are derived from OISST [35] available at a resolution of 0.25° × 0.25° since 1982. Data are available at: https://psl.noaa.gov/data/gridded/data.noaa.oisst.v2.html. Before that, we used gridded monthly HadISST data [36] at 1° × 1° resolution over the period 1948 to 1982, which are available at: https://www.metoffice.gov.uk/hadobs/hadisst/data/download.html.

The location and areal extent of the Ganges and Brahmaputra river basins are obtained from the “Total Runoff Integrating Pathways” (TRIP) [37] database at 0.5 × 0.5° spatial resolution available at: http://hydro.iis.u-tokyo.ac.jp/~taikan/TRIPDATA/TRIPDATA.html.

### 2.2. Satellite Water Marker

Satellite-measured radiance data are used to calculate the Satellite Water Marker (SWM) index proposed by Jutla et al. (2013) [3]. Specifically, the SWM is using the ratio between measures of water turbidity and water clarity at remote sensing reflectance (*Rrs*) wavelengths of 555 nm (green yellow) and 412 nm (purple blue) respectively:


(1)
$$SWM=\left[\frac{R_{rs}\left(555\right)-R_{rs}\left(412\right)}{R_{rs}\left(555\right)+R_{rs}\left(412\right)}\right]\times 100.$$


The remote-sensing reflectance of *Rrs* 412 nm and *Rrs* 555 nm are obtained from the European Space Agency Ocean Colour Climate Change Initiative (ESA OC-CCI) [38] at monthly temporal resolution and 4 × 4 km spatial resolution over the period 1997–2018 available at: https://climate.esa.int/en/projects/ocean-colour/data/ The SWM data were regridded to 0.25° × 0.25° spatial resolution by averaging all available data points within each new, larger pixel.

### 2.3. Statistical Methods

The correlation analyses presented in the manuscript are based on the period 1997 to 2016, spanning the availability of both climate and satellite water marker datasets. We calculated and plotted results in all figures except Figure 4 (Pearson correlation coefficients, composite maps, standard deviation) using the Grid Analysis and Display System (GrADS) version 2.0 and 2.1 (http://cola.gmu.edu/grads/). Regarding the linear regression model in Figure 4, we calculated it using the Microsoft Excel package.

## 3. Results

### 3.1. Climate Precursors of Fall SWM in Northern Bay of Bengal

We calculated lagged correlation between anomalies of SST, rainfall, and SWM October–December (OND) index, which has been shown as a precursor of spring cholera outbreak over northern BoB (20–23° N, 85–95° E) [3]. The SST correlation analyses show a significant La Niña pattern over the tropical Pacific in fall (OND; Figure 1a), which can be traced back to the previous season in the summer monsoon (July–September, JJAS; Figure 1b). In addition, an off-equatorial SST signal over the northeastern Pacific off California was observed two seasons before (in spring, pre-monsoon March–May, MAM; Figure 1c). The latter SST anomaly distribution resembles that of the Pacific Meridional Mode, which is known as a precursor of ENSO [39,40]. Previous studies have suggested that such a La Niña-like SST pattern contributes positive rainfall anomaly [41,42], which is expected to affect the coastal water conditions over the northern BoB.

The rainfall correlation analyses show that during the summer Asian monsoon season, a positive rainfall anomaly appears over northern India around 20–30° N, 75–90° E (JJAS; Figure 1e). Excessive rainfall, coming from the Ganges river basin (i.e., the area shaded in dark-gray color in Figure 1e), causes flooding and an increase of river discharges over the Bengal Delta. The turbid water pouring from the Bengal Delta is expected to create favorable coastal-habitat conditions for the *V. cholerae* bacteria. In contrast, the same season correlation analysis between fall SWM over the northern BoB and fall rainfall over northern India (area average over 20–30° N, 75–90° E) reveals a negative rainfall anomaly over central India (Figure 1d). This result indicates that the fall rainfall anomaly over India is not responsible for changes in the river discharge and fall SWM over the Bengal Delta. Furthermore, a recent study with a climate prediction system based on a ocean–atmosphere coupled general circulation model [33] has shown a 2-year prediction skill for Indian summer monsoon rainfall, which is primarily driven by the multi-year memory of a tropical ocean–atmosphere coupled system such as ENSO. The results presented in the present paper and shown in Figure 1 also imply the SWM predictability beyond the season through the ENSO–monsoon relationship. This ENSO–monsoon relationship is discussed in Section 3.2 and Section 3.5.

### 3.2. Climate Precursors of North India Summer Monsoon Rainfall

In addition to influencing the fall SWM index and associated spring cholera (as shown above), summer monsoon rainfall is an important factor for the fall cholera outbreak through flood-induced contamination of inland and coastal waters associated with perturbations on socioeconomic conditions, including hygiene practices, access to safe drinking water, and the availability of sanitation and drainage systems [7,15].

To investigate the connection to large-scale patterns of climate variations, we calculated the lagged correlation between the north India Summer monsoon rainfall (JJAS) and global SST anomalies. A significant La Niña pattern over the tropical Pacific during the summer monsoon months (JJAS; Figure 2a) suggests a link between monsoon rain and the ENSO. This signal can be traced back to the previous spring pre-monsoon season (MAM; Figure 2b). Furthermore, two seasons before, the off-equatorial SST signal over the northeastern Pacific coast of California (JF; Figure 2c) resembles the Pacific Meridional Mode seen as a precursor of ENSO. These SST features are generally similar to the patterns in Figure 1. A La Niña-like SST pattern is expected to affect the river discharge anomaly over the Bengal Delta region through the above normal monsoon rainfall anomaly over northern India. Furthermore, during this season, a positive rainfall correlation is observed over the northern Indian region (Figure 2d) as anticipated based on a rainfall index taken from that region.

### 3.3. Mechanistic Link between Climate and Seasonal Cholera in Bay of Bengal Coastal Region

Based on the results of the analyses of the spring outbreak conditions and SWM (Figure 1) and Indian monsoon rainfall (Figure 2), we summarized the climate precursors for spring and fall cholera risk in the north BoB coastal region (Figure 3). First, we established that summer rainfall variability over northern India is an important driving factor of the variability of the fall SWM, which has been identified by previous study [3] (Jutla et al. 2013) as a good predictor of cholera prevalence during the following spring months through the the intrusion of coastal waters with suitable conditions for *V. cholerae* bacteria (Figure 3, steps 1, 2, 3a, and 4). The high SWM anomaly is correlated with northern Indian rainfall and La Niña-like SST anomalies in summer monsoon (JJAS), and it can be further traced back to one or two seasons before, in spring (MAM) or winter (JF) respectively, based on SST precursors in eastern Pacific and Pacific meridional mode.

Second, we elucidated the climate precursors of the summer monsoon rainfall and associated fall cholera (Figure 3, steps 1, 2, and 3b). Based on correlation analyses, we observed that La Niña or Pacific meridional mode SST anomaly one or two seasons before, in the spring (MAM) or winter (JF), respectively, can be conducive to high summer rainfall anomaly over north India. Excess rainfall in this region leads to flood and increased river discharge over the Bengal Delta, which are associated with higher water turbidity and increased risk of water contamination through perturbations of socioeconomic factors: notably, damages to drinking water and sanitation infrastructures.

### 3.4. SWM Prediction Model Using Climate Precursors

As fall SWM is a good indicator for *V. cholerae* habitat suitability and the subsequent prevalence of cholera in spring, we examined SWM interannual variability in relation to rainfall. The fall SWM anomaly over the northern BoB (blue lines in Figure 4) are positive in 1999, 2003, 2007, 2008, 2011, and, 2016, and negative in 1997, 2001, 2006, 2009, 2012, 2014, and 2015. This anomaly is shown to be positively correlated with the rainfall anomaly over northern India, particularly for the later monsoon season (Figure 4a, R = 0.58) of August–September (AS). The fall SWM anomaly is also negatively correlated with the Nino 3.4 index in JJAS (Figure 4b, R = −0.59), suggesting that the summer conditions of Nino 3.4 and rainfall over northern India are good precursors of the SWM anomaly during fall/early winter (OND). In addition, the Indian Ocean Dipole Mode Index (DMI) has previously been reported to contribute the southern Asian monsoon variability. Based on these results, we constructed a simple regression model as follows:SWM=α×Nino3.4+β×Rain+γ×DMI.

The fall SWM anomaly estimated by the multi-linear regression model (Figure 4c) shows higher correlation (R = 0.71) when including Niño 3.4, DMI, and rainfall compared to linear models that included only rainfall (R = 0.58) or only Nino 3.4 (R = −0.59) anomalies. Including DMI contributes a slightly higher correlation coefficient compared to a multi-linear model without DMI anomaly (Figure 4d, R = 0.68). The correlation coefficients and R^2^ values for these models are summarized in Figure 4.

### 3.5. Revisit of Decadal Changes in Monsoon–ENSO/DMI Relationship

Correlation analysis between northern Indian summer monsoon and ENSO shows a significant relationship (R = −0.48) over the period 1948–2020. The negative correlation coefficient means that rainfall increases (decreases) during La Niña (El Niño), which is consistent with the results presented in Figure 1 and Figure 2. However, such monsoon–ENSO relationships can vary over an interdecadal time scale [43,44]. For example, the monsoon–ENSO relationship is found to be significant during the periods 1950–1977 and after 2005 but non-significant in the period 1977–2005 (Figure 5). In addition to the monsoon–ENSO relationship, the DMI is further considered as a key factor of south Asian monsoon variability [43,45,46]. Here, we find that the correlation between the DMI and the rainfall variability over northern India becomes significant in the periods 1977–1990 and 2004–2010 (Figure 5). Such an interdecadal change of the monsoon–ENSO/DMI relationship suggests that the statistical fitting of the fall SWM variability presented in Figure 4 may be non-stationary on interdecadal time scales. It is interesting to note that two periods with a significant monsoon–ENSO relationship (between 1950 and 1977 and after 2005) are both in the negative phase of the Pacific Decadal Oscillation. This implies that the decadal variability may contribute to the changing of the monsoon–ENSO relationship through the background state change, and this should be investigated further in future study.

To revisit the decadal changes in the monsoon–ENSO/DMI relationship from the perspective of the climate precursor of the SWM index, we explore lagged relationships between the Indian monsoon rainfall and ENSO in previous seasons. We define three decadal periods based on the correlation with summer rainfall identified in Figure 5: 1982–1991 (significant correlation for DMI but non-significant for Nino 3.4; see Figure 5), 1992–2001 (non-significant correlation for both DMI and Nino 3.4), and 2002–2011 (significant correlation for both DMI and Nino 3.4). Next, we produced maps of the correlation between August–September rainfall anomaly over northern India and the SST anomaly for these three periods (Figure 6).

In the periods of 1982–1991 and 1992–2001, the positive rainfall anomaly over northern India shows a significant correlation with El Niño-like warm SST anomaly in the previous winter (Figure 6a,d), but this positive SST correlation pattern decays in the following seasons, and the correlation becomes non-significant (Figure 6c,f). Contrastingly, during the period of 2002–2011, the correlation between rainfall over northern India and tropical SST anomaly is markedly different (Figure 6g–i). The El Niño-like positive correlation cannot be seen in the previous winter (Figure 6g), and rather, a La Niña-like negative correlation (northern India rainfall increases with SST cooling in the tropical Pacific) is observed (Figure 6i).

The significant correlation of El Niño-like warm SST anomalies in winter and spring with the summer rain index (AS), except for the 2002–2011 period, is due to the so-called “capacitor effect” [47] through which a strong El Niño event in the previous winter causes Indian Ocean basin warming in the following spring and anomalous tropospheric warming over the tropical Indian Ocean and anticyclonic response over the western North Pacific in the following boreal summer season. These delayed responses to strong El Niño events in the previous winter may improve the potential predictability of Asian monsoon variability beyond the season [33,44].

Previous climate studies have established that the Indian monsoon rainfall increases (decreases) during La Niña (El Niño) based on the monsoon–ENSO/DMI relationship and further revealed interdecadal changes in this relationship [44,48,49]. However, a possible link and impact on cholera prevalence on an interdecadal time scale is a new finding owing to the availability of a new long-term continuous dataset of SWM index produced in the present study for the entire northern BoB region at high spatial resolution and high repeat frequency using climate-quality satellite observations. Interdecadal changes in the correlation between south Asian summer monsoon rainfall and climate modes indicate that the ENSO and cholera outbreak relationship may be unstable over a decadal time scale and underline the difficulty of using climate precursors for long-term analysis and prediction of seasonal cholera in the north BoB coastal region. For example, using composite analysis, we find a marked ENSO–monsoon influence on coastal SWM over the northern region of BoB, La Niña, and summer flood in the north of India during the period 2006–2017 when the ENSO–monsoon relationship is significant (Figure 5 and Figure 7). In contrast, the ENSO–monsoon influence is not observed in the period 1997–2005 when the ENSO–monsoon relationship is non-significant (Figure 5 and Figure 8).

For the fall cholera outbreaks, a relationship with winter El Niño through summer flooding over northern India has been previously suggested [28]. These results are consistent with the findings presented in Figure 6, showing winter El Niño in 1988 and 1998, which corresponds to the years when fall cholera outbreaks have been reported as a flood-associated epidemic [7]. One exception is the fall cholera outbreak in 2004, which did not occur after a winter El Niño but was related to summer flooding driven by the boreal summer intra-seasonal oscillation (BSISO) [50,51].

## 4. Discussion

Previous studies have revealed that satellite monitoring of chlorophyll-a concentration and SWM index is a useful tool of favorable conditions for *V. cholerae* in coastal areas [3,30], but we investigated that the SWM index can be predictable using climate variables (i.e., SST and rainfall) two seasons earlier. In the Bengal Delta, river discharge plays an important role for the chlorophyll-a concentration, which is essentially different from the coastal upwelling region such as the Sumatra–Java coast [31,32]. In addition, salinity is also important for the growth of *V. cholera* [11]. To improve the predictability, further studies using salinity datasets will be needed.

Furthermore, recent global warming Earth system model simulations have reported an increase in strong positive DMI event in a warmer climate [52], which could lead to more frequent climate extremes in the region and increased contamination risk from flooding and the prevalence of favorable conditions for *V. cholerae* in coastal waters.

## 5. Conclusions

In this study, using satellite and climate datasets, a mechanistic link between climate variability and cholera (Satellite Water Marker; SWM) index in the Bengal Delta was demonstrated. The mechanistic link allows us to find precursors of cholera outbreaks up to two seasons earlier (Figure 1). High values of the SWM index in fall were associated with above-normal summer monsoon rainfalls over northern India. In turn, these correlated with the La Niña climate pattern that was traced back to the summer monsoon and previous spring seasons (Figure 2). We presented a new multi-linear regression model (Figure 4) that can explain 50% of the SWM variability over the Bengal Delta based on the relationship with climatic indices of Niño 3.4, DMI, and summer monsoon rainfall during the decades 1997–2016. Interestingly, we further found that these relationships were non-stationary over the multi-decadal period 1948–2018 (Figure 5, Figure 7 and Figure 8). These results bear novel implications for developing outbreak-risk forecasts, demonstrating a crucial need to account for multi-decadal variations in climate interactions and underscoring to better understand how the south Asian summer monsoon responds to climate variability.

## Figures and Tables

**Figure 1 ijerph-18-10201-f001:**
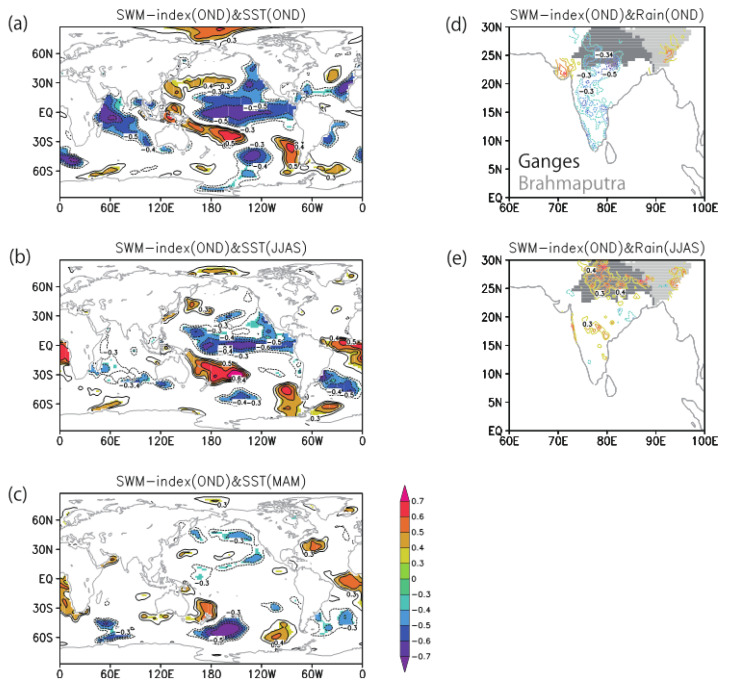
Climate precursors of fall SWM index over the north Bay of Bengal. Correlation maps between the October and December (OND) SWM index over the north BoB (20–23° N, 85–95° E) and (**a**) same-season OND SST anomaly, (**b**) one-season lead June–September (JJAS) SST anomaly, (**c**) two-season lead March–May (MAM) SST anomaly, (**d**) same-season OND rainfall anomaly, and (**e**) one-season lead JJAS rainfall anomaly. In (**a**–**c**), only areas above 90% significance are shaded. In (**d**,**e**), contour intervals are 0.1 (less than ±0.3 are omitted).

**Figure 2 ijerph-18-10201-f002:**
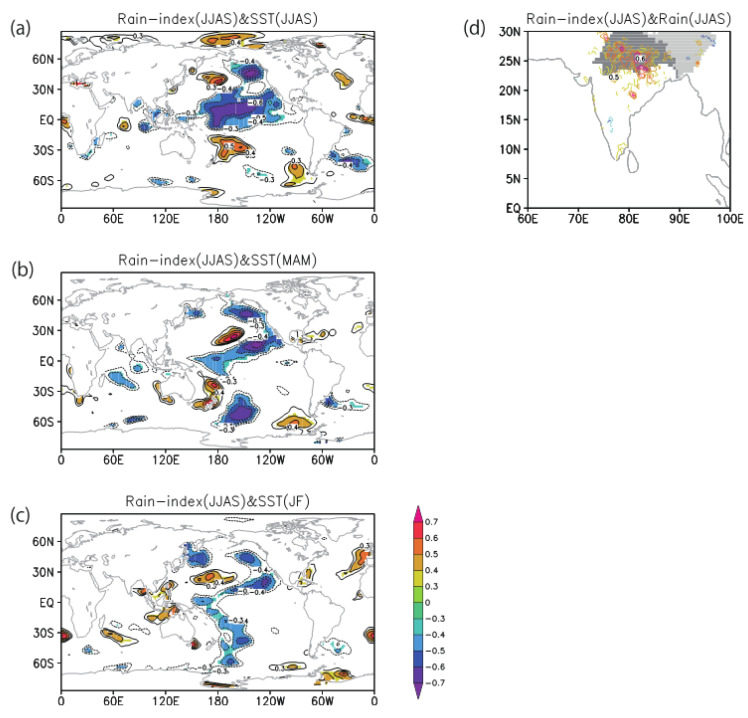
Climate precursors of north India summer monsoon rainfall. Correlation map between June–September (JJAS) rainfall variability over northern India (20–30° N, 75–90° E) and (**a**) same season JJAS SST anomaly, (**b**) one-season lead March–May (MAM) SST anomaly, (**c**) two-season lead January–February (JF) SST anomaly, and (**d**) same-season JJAS rainfall anomaly. In all panels, only areas above 90% significance are shown.

**Figure 3 ijerph-18-10201-f003:**
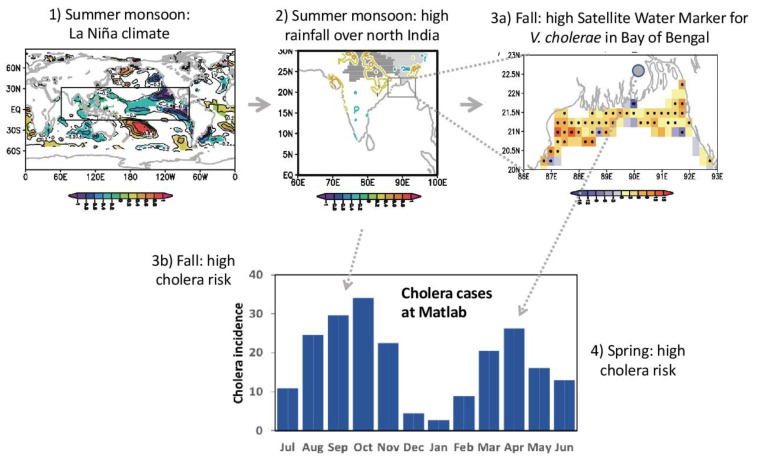
Mechanistic link between climate conditions and seasonal cholera in Bay of Bengal coastal region. In steps (1), (2), and (3a), composite maps are defined as high years (exceeding ±1σ) of the October–December (OND) SWM index over the north BoB (20–23° N, 85–95° E). (1) June–September SST anomaly, (2) normalized August–September rainfall anomaly (only ±1σ, ±1.5σ and above are contoured), (3a) SWM anomalies in OND. Only areas above 90% significance are shown. Composite maps are produced over the period 1997–2016. In steps (3b) and (4), monthly average number of cholera cases recorded at Matlab (a district of Bangladesh) over the period 1998–2007. The cholera data have been digitized from Akanda et al. (2009, their Figure 1a) using WebPlotDigitizer software (https://automeris.io/WebPlotDigitizer/, accessed on 2 March 2021).

**Figure 4 ijerph-18-10201-f004:**
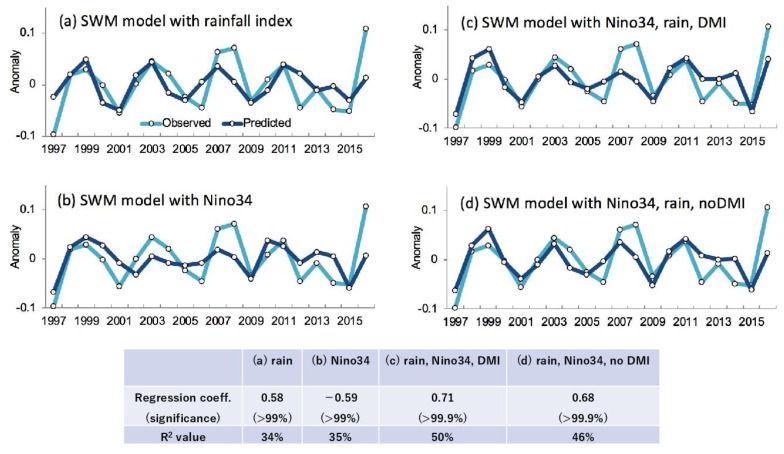
Regression model to predict SWM over the north BoB. Time series of fall SWM anomaly over the north BoB (20–23° N, 85–95° E) over 1997–2016: observed (blue curves) and predicted (purple curves) using a regression model based on (**a**) August–Sepetember (AS) rainfall anomaly over north India (20–30° N, 75–90° E), (**b**) June–September (JJAS) Nino 3.4 anomaly, (**c**) AS rainfall, JJAS Nino 3.4 anomalies, and JJAS Dipole Mode Index (DMI), and (**d**) AS rainfall, JJAS Nino-3.4, no DMI anomalies.

**Figure 5 ijerph-18-10201-f005:**
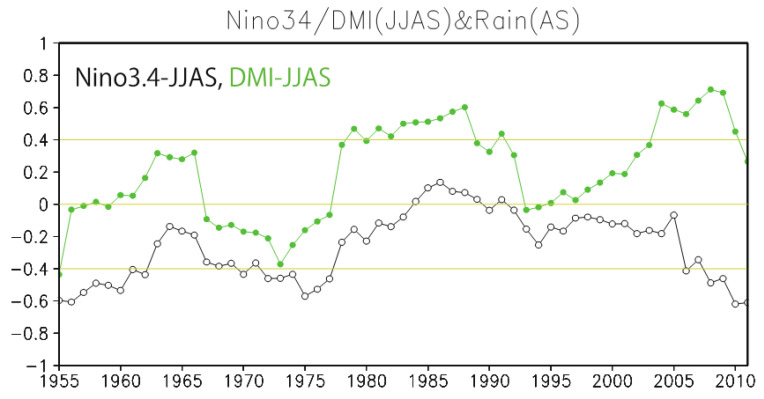
Interdecadal changes in monsoon–ENSO/DMI relationships. Eleven-year sliding correlation between August–September (AS) rainfall anomaly over northern India (20–30° N, 75–90° E) and June–September (JJAS) Nino 3.4 (black) and Dipole Mode Index (DMI) (green) indices from 1948 to present.

**Figure 6 ijerph-18-10201-f006:**
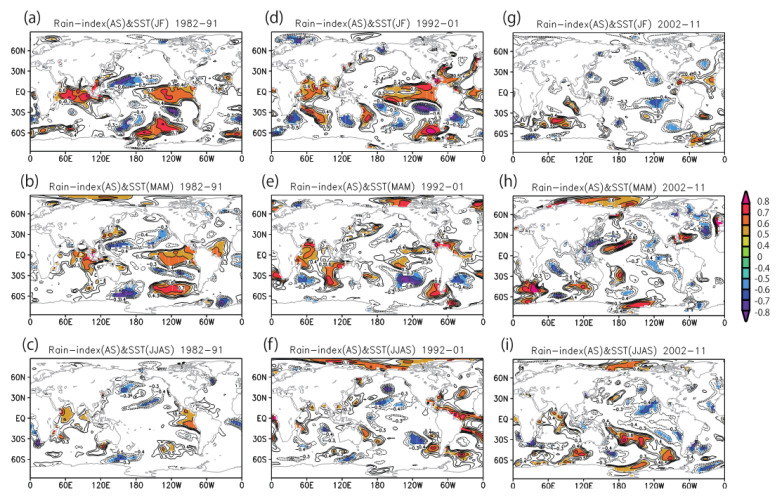
Changing relationship of lagged correlation between SST and rainfall over northern India. Correlation maps between August–September (AS) rainfall variability over northern India (20–30° N, 75–90° E) and (**a**) two-season lead January–February (JF) SST anomaly, (**b**) one-season lead March–May (MAM) SST anomaly, and (**c**) simultaneous June–September (JJAS) SST anomaly during 1982–1991. (**d**–**f**) same as (**a**–**c**) but for 1992–2001. (**g**–**i**) same as (**a**–**c**) but for 2002–2011. Only areas above 90% significance are shown. Contour intervals are 0.1 (less than ±0.3 are omitted).

**Figure 7 ijerph-18-10201-f007:**
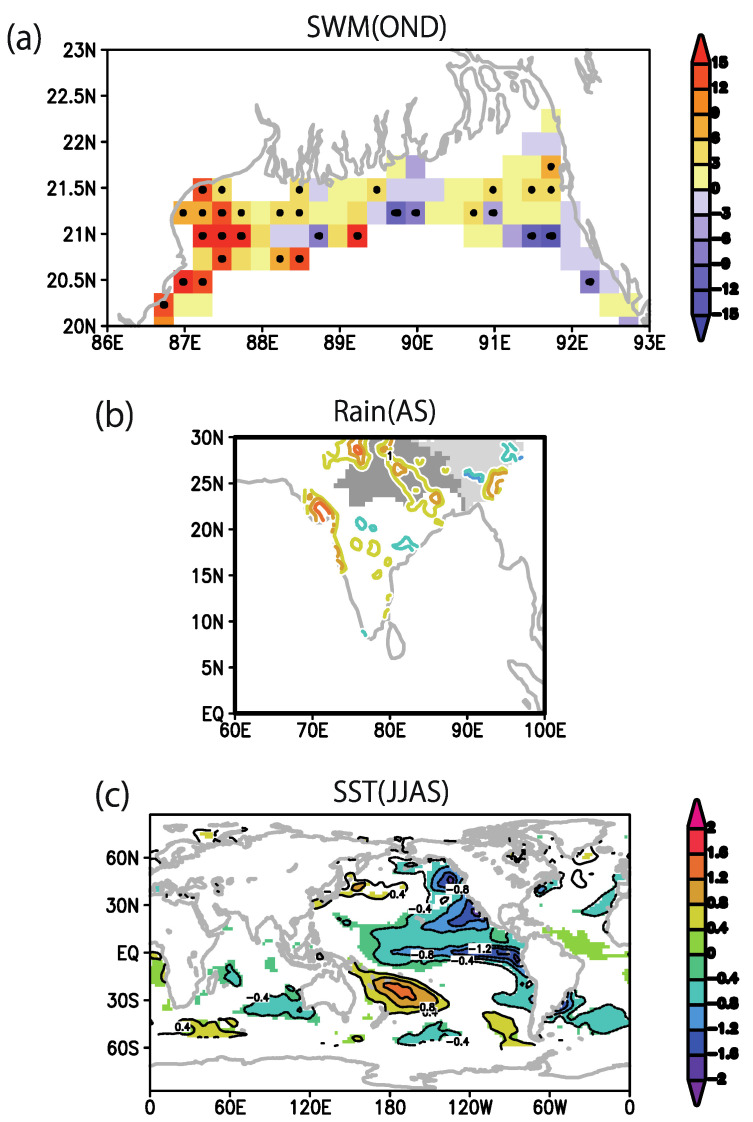
Composite map of high-Satellite Water Marker (SWM) over northern BoB in 2006–2017 (period of significant ENSO-monsoon relationship, see Figure 5). Composite map defined by high and low (exceeding ±1σ) years of October–December (OND) SWM index over the north BoB (20–23° N, 85–95° E). (**a**) SWM anomalies in OND, (**b**) normalized August–September (AS) rainfall anomaly (only ±1σ, ±1.5σ … are contoured), and (**c**) same season June–September (JJAS) SST anomaly. Only areas above 90% significance are shown.

**Figure 8 ijerph-18-10201-f008:**
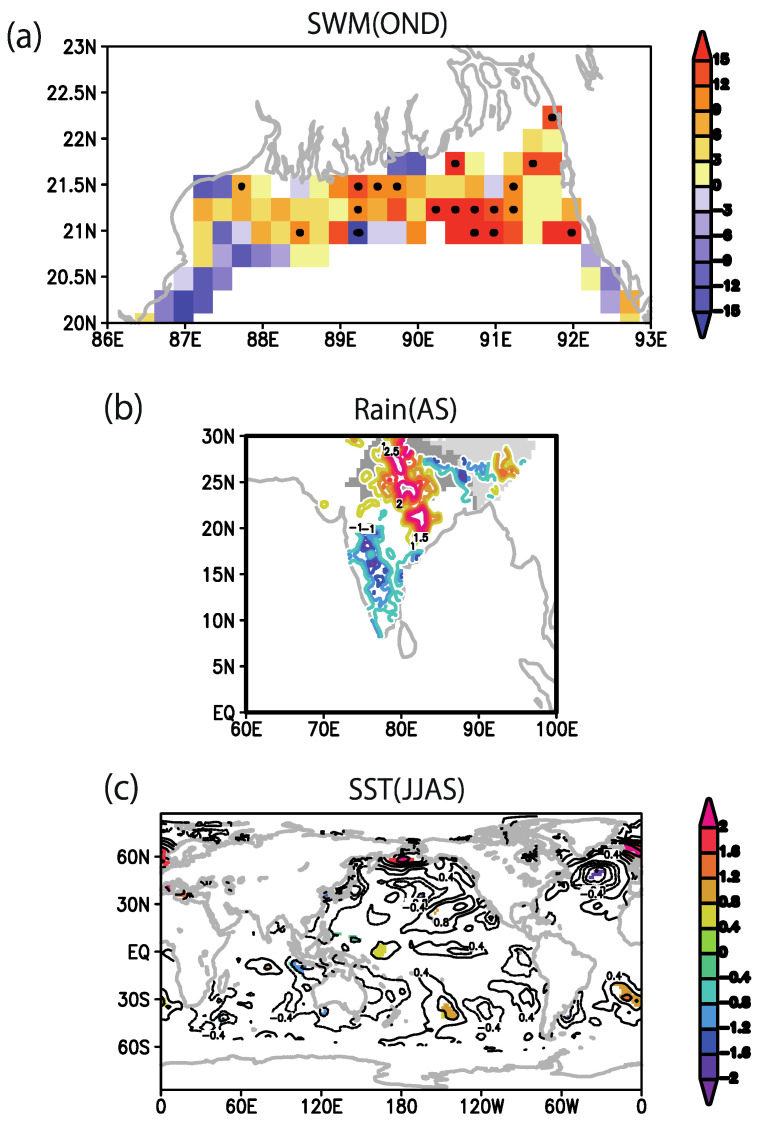
Composite map of high-Satellite Water Marker (SWM) in 1997–2005 (period of insignificant ENSO–monsoon relationship; see Figure 5). Composite maps defined by high and low (exceeding ±1σ) years of October–December (OND) SWM index over the north BoB (20–23° N, 85–95° E). (**a**) SWM anomalies in OND, (**b**) normalized August–September (AS) rainfall anomaly (only ±1σ, ±1.5σ … are contoured), and (**c**) same-season June–September (JJAS) SST anomaly. Only areas above 90% significance are shown.

## Data Availability

The model data used in this study are available upon request. If data are needed, please contact the corresponding author (ogatatom@jamstec.go.jp).

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
