# Peer review of "Climate Precursors of Satellite Water Marker Index for Spring Cholera Outbreak in Northern Bay of Bengal Coastal Regions"

_ijerph, 2021, doi:10.3390/ijerph181910201_

Round 1

Reviewer 1 Report

This manuscript authored by Ogata et al, entitled ‘Climate precursors of satellite water marker index for spring cholera outbreak in northern Bay of Bengal coastal regions’, presents an interesting study linking large-scale climate dynamics with regional rainfalls in the Bay of Bengal and associated outbreaks of the cholera disease, focus on annual and interannual scales. Although I do not know infectious diseases and public health better than ordinary people, it is obvious that the accurate early prediction of the outbreaks of infectious diseases is of broad interest. From the perspective of a climatic scientist, the manuscript is well-prepared. The analysis is solid and reliable. The discussion is informative and helpful. The findings are important.

In my humble opinion, the manuscript can be published with some minor revisions. I only have some minor questions, see below.

Specific Questions:

Line 172-173: Prediction from what variable? Should clarify.

Figure 1, Section 3.4. etc: if the cholera outbreak data are available, why not correlate that with SST and rain directly? In other words, it seems that the SWM index as a proxy is unnecessary here. Could a prediction model for the cholera outbreak itself be derived? Justification required.

Line 266-268: This phenomenon is interesting. What controls this different relationship? Except for DMI, it appears that two periods with a significant monsoon-ENSO relationship (between 1950-1977 and after 2005) are both in Pacific Decadal Oscillation (PDO) negative phase. Could PDO be a factor? I would like to see some discussions about this point.

Figure 3: what is the meaning of Matlab in 3b) here?

Figure 6: It surprised me since Figure 6 (a-f) all correspond to an uncorrelated Nino3.4, yet SST in preceding seasons (JF and MAM) all show strong positive correlations in the tropical oceans (Indian and Pacific). What if the authors correlate Nino3.4 with two-season leads with the summer rain index (AS), excluding the 2002-2011 period?

Author Response

Please see the attached docx file.

Reviewer 2 Report

I think that the manuscript is key for the prediction of cholera epidemics in endemic regions and their relation with global acting phenomena.

The discussion and the conclusions are very short, you should extend them comparing it with other studies working with chlorophyll or plankton, or with other environmental factors such as salinity. You work with turbidity and an important component of it is the organic matter, may be you should discuss the role of the organic matter for V. cholerae.

Minor commentaries:

Line 14) exclude for the first time, check the references

Line 50) include also sewage

Line 90) season

Author Response

Please see the attached docx file.
